# Nitrogen Supplementation Modulates Morphological, Biochemical, Yield and Quality Attributes of Peppermint

**DOI:** 10.3390/plants12040809

**Published:** 2023-02-10

**Authors:** Zubair Ahmad Parrey, Sajad Hussain Shah, Mudasir Fayaz, Ryan Casini, Hosam O. Elansary, Firoz Mohammad

**Affiliations:** 1Plant Physiology and Biochemistry Section, Department of Botany, Aligarh Muslim University, Aligarh 202002, Uttar Pradesh, India; 2Plant Tissue Culture Research Laboratory, Department of Botany, University of Kashmir, Srinagar 190006, Jammu and Kashmir, India; 3School of Public Health, University of California, Berkeley, 2121 Berkeley Way, Berkeley, CA 94704, USA; 4Department of Plant Production, College of Food & Agriculture Sciences, King Saud University, P.O. Box 2460, Riyadh 11451, Saudi Arabia

**Keywords:** growth, microscopy, nitrogen, peppermint, physio biochemistry, yield and quality

## Abstract

Due to the rising demand for essential oil in the world market, peppermint has gained an important status among aromatic and medicinal plants. It becomes imperative to optimize its performance in terms of the growth, physiological functioning and biosynthesis of specialized metabolites. A factorial randomized pot experiment was performed using three peppermint cultivars (Kukrail, Pranjal and Tushar) and five levels of leaf-applied nitrogen (N), viz. 0 (control), 0.5, 1.0, 1.5 and 2%. The phenological features, biochemical parameters, viability of root cells, stomatal and trichome behavior were assessed at 100 days after transplanting (DAT). The yield-related parameters, viz., herbage yield, essential oil content, menthol content and yield were studied at 120 DAT. The results revealed that increasing the N doses up to 1.5% enhanced all the studied parameters of peppermint, which thereafter (at the dose above 1.5% N) decreased. The variation pattern of the studied parameters was “low-high-low”. Cultivar Kukrail surpassed the two other cultivars Tushar and Pranjal. Among the foliar sprays, the application of 1.5% N increased chlorophyll content and net photosynthetic rate in all three cultivars. Moreover, the essential oil (EO), EO yield and menthol yield of the plant were also increased linearly in all three cultivars as compared with their control plants. Nitrogen application enhanced the trichome size and density of the plants, as revealed through scanning electron microscopy. Furthermore, from the GC-MS studies, the EO content in the studied cultivars increased, particularly in the case of menthol, with the N application. It may be concluded that two sprays of N (1.5%) at appropriate growth stages could be beneficial for improving morphological, physio biochemical and yield attributes of peppermint.

## 1. Introduction

Peppermint (*Mentha piperita* L.) is a strongly scented, perennial herbaceous plant of the family Lamiaceae. Peppermint is an important source of medicinal and aromatic compounds and has achieved an important status among the medicinal and aromatic plants due to the presence of bioactive compounds such as menthol, menthyl acetate, menthone and menthofuran, etc. It is widely employed for curing various health-related ailments as an analgesic, antiseptic, carminative, decongestant, stomachic and stimulant among other uses [1,2]. Moreover, its essential oil (EO) content has been well known to have many properties including antibacterial, antifungal, antimicrobial, antiallergic, antihypertensive, antiviral, anti-inflammatory and antioxidant [3]. The global demand for its plant-based specialized metabolic compounds requires its cultivation worldwide. Plants require mineral elements for their optimum growth and development [4,5]. Nitrogen (N) is one of such essential elements and is a constituent of many key cellular functional compounds such as amino acids, proteins, photosynthetic pigments, enzymes, co-enzymes and nitrogenous bases in plants [6,7]. It regulates plant growth, development and metabolic activity and is required in large quantities [8]. The soil-applied N is lost due to volatilization, leaching, fixation decomposition by microorganisms and other factors and does not remain fully available to plants. Further, it negatively impacts the soil microbiome which plays a significant role in N cycling. Limited N availability in the soil results in improper plant functioning of the growing plant, for example, by impacting photosynthetic CO_2_ assimilation, photosynthetic rate, quantum yield and Ribulose-1,5-bisphosphate carboxylase/oxygenase (RuBisCo) activity to a great extent [9,10]. Therefore, N plays a substantial role in stimulating the metabolic and developmental processes of plants during their entire life cycle.

Foliar feeding is considered an effective and better option for providing nutrients to plants quickly during vegetative and reproductive phases than soil fertilization. The foliar application of N is one of the improved N management practices with minimal N loss to meet its requirement efficiently. The exogenous N is predominantly applied in the form of urea to plants and is rapidly absorbed from the surface of leaves with the help of specialized transporters such as degradation of urea (DUR3) and tonoplast intrinsic protein (TIP); it is hydrolyzed by urease into ammonia which in turn is assimilated into amino acids [11,12]. Therefore, the foliar application of N reduces the wastage of N and increases the N-use efficiency on a large scale which in turn minimizes the undesirable repercussions on soil, water and air quality [13]. Previous studies have also reported that the foliar application of N improved growth and development, photosynthetic rate and secondary metabolites in plants in comparatively lower quantities [14]. To the best of our knowledge, no research has been yet performed elucidating the effect of N supplementation on phytochemical yield and content relating to cellular viability, stomatal and trichome behavior in peppermint.

Therefore, with the backdrop of our involvement with enhancing plant nutritional qualities [5,15], this experiment was devised to investigate the effect of exogenously applied N on the performance of peppermint in terms of growth, physiological characteristics, and EO biosynthesis besides evaluating cellular viability, stomatal and trichome behavior. Moreover, as different cultivars are known to differ in their mineral nutritional requirement pertaining to varied biochemical and morpho-physiological characteristics, the effect of N was studied comparatively on three different cultivars of peppermint.

## 2. Results

### 2.1. Nitrogen Supplementation Modulates the Morphology of Peppermint

The exogenous application of N was found to enhance the growth characteristics of all three cultivars significantly. It was observed that with an increase in % N supplementation, the growth attributes were enhanced by increased N levels up to 1.5%N. In all the three cultivars 1.5% N treatment was found optimum for all the morphological parameters studied. At 1.5% N supplementation Kukrail showed the highest increase of 36.03% in shoot length, 34.33% in root length, 23.03% in the area of leaf and 72.86% in leaf area of the plant; Pranjal showed the least effects with enhancement of 23.52% in shoot length, 23.09% in root length, 18.59% in area per leaf and 48.75% in leaf area per plant over respective controls (Figure 1). The other parameters such as leaf number per plant, shoot fresh weight, root fresh weight, shoot dry weight and root dry weight per plant were also increased by 23.03 and 18.59%, 28.80 and 20.10%, 33.08 and by 19.88%, 22.42 and 18.59%, 27.60 and 12.71% in Kukrail and Pranjal cultivars, respectively, compared to the control plants (Figure 2). Shoot and root length per plant and leaf area per plant showed the maximum variance among the cultivars with respect to treatments (Figure 1 and Figure 2).

### 2.2. Nitrogen Supplementation Regulates Physio-Biochemical Attributes and Mineral Nutrients of Peppermint

Cultivar Kukrail responded better among the three cultivars surpassing the Tushar and Pranjal cultivars. The foliar application of 1.5% N enhanced chlorophyll content by 33.50 and 19.32%, *P_N_* by 39.08 and 22.03%, *gs* by 39.28 and 24.09% *E* by 17.36 and 14.81%, *Ci* by 12.10 and 10.35%, CA activity by 24.62 and 22.33% in Kukrail and Pranjal, respectively, over their respective controls (Figure 3A–F). Similarly, the cultivar Kukrail surpassed the other cultivars in leaf mineral nutrient content, followed by Tushar and Pranjal. The leaf N content was increased by 60.17 and 43.98% in Kukrail and Pranjal, respectively, with the highest variability. The leaf P content was increased by 34.3 and 17.24% and leaf K content by 13.93 and 10.0% in cultivar Kukrail and Pranjal, respectively, over their respective control plants (Figure 4A–C).

### 2.3. Effect of N Application on the Yield and Quality Attributes of Peppermint

The foliar treatment with N influenced the yield parameters of all studied cultivars. The graded levels of N up to 1.5% N improved the yield attributes which thereafter decreased at 2% N. The cultivar Kukrail responded well and was accompanied by Tushar and Pranjal. The herbage yield per plant was increased by 27.35 and 21.36%, EO content by 34.42 and 20.37%, and EO yield per plant by 73.17 and 43.75% in Kukrail and Pranjal, respectively, over their respective controls (Figure 4D–F).

#### Effect of N on the Menthol Content of Peppermint Cultivars

The GC-MS technique was used to analyze and compare the bioactive compounds of three studied cultivars. However, only menthol content was further analyzed in terms of N supplementation. At 1.5% N, the menthol content of the Kukrail cultivar increased by 1.38 times that of the controls, followed by the Tushar and Pranjal cultivars at 1.34 and 1.26 times, respectively, over their respective controls (Figure 5A–C). The menthol yield was increased by 150 and 85.71% in Kukrail and Pranjal, respectively, over their control (Figure 5D). The representative GC-MS chromatograms of the Kukrail (best performer) cultivar is given in Figure 6.

### 2.4. Microscopical Analysis

#### 2.4.1. Confocal Laser Scanning Microscopy

A red-fluorescent intercalating dye known as propidium iodide (PI) was used to stain cells that bind with genetic material in order to visualize cell viability. The PI is cell-impermeable and so cannot enter live cells. It enters into the dead cells via damaged portions of the cell membrane and is then intercalated between the base pairs of deoxyribonucleic acid (DNA), resulting in red fluorescent spots. In our study, the three cultivars receiving 1.5% N spray treatment showed the least fluorescence during microscopic analysis (the samples taken from the different treatments of individual cultivars were compared as well as comparison between the cultivars). On the basis of observation, the maximum viability of root cells was reported at 1.5% N application when compared with their respective controls. However, the highest viability of cells was confirmed at 1.5% N treatment in the cultivar Kukrail surpassing the cultivars Tushar and Pranjal (Figure 7).

#### 2.4.2. Scanning Electron Microscopy

Stomata and trichomes were examined using SEM and then analyzed with ImageJ software to determine their size and density. The SEM analysis revealed that the foliar treatment of 1.5% N increased the stomatal dimensions (length and width) and density, as well as the size and density of trichomes on the leaf surface. Kukrail, followed by Tushar and Pranjal, had the best values for these parameters. According to the SEM analysis, the observed mean ± SE of stomatal aperture length and stomatal aperture width of the Kukrail cultivar treated with 1.5% N was 15.24 ± 0.28 µm and 2.54 ± 0.85 µm, respectively, and the mean ± SE of stomatal density was 62 ± 2.25 mm^2^, whereas in its control plants, the mean ± SE of stomatal aperture length and stomatal aperture width were 10.45 ± 1.69 µm and 1.67 ± 0.24 µm, respectively, and the mean ± SE of stomatal density was 45 ± 1.42 mm^2^. The measured mean ± SE of trichome size and density of the Kukrail cultivar at 1.5% N was 75.54 ± 2.85 µm and 9 ± 0.12 mm^2^, whereas the mean ± SE of trichome size and density of its control was 66.54 ± 2.17 µm and 4 ± 0.11 mm^2^ respectively (Figure 8A–E). The representative micrographs are displayed in Figure 9, Figure 10, Figure 11 and Figure 12.

### 2.5. Principal Component and Heat Map Analysis

According to principal component analysis, 85.2% of the variance can be attributed to principal component 1 (Dim 1), whereas 4.5% of the variance can be attributed to principal component 2 (Dim 2). The impact of N supplementation on all of the measured variables was analyzed using principal components. The variance in the studied parameters was evaluated in terms of cultivars (Figure 13) and treatments (Figure 14).

According to the PCA-biplot (Figure 13), the cultivar Pranjal has the most variance when compared to Kukrail and Tushar. Despite this, the score plot separated the Tushar and Kukrail cultivars; however, many of the variables overlap. Dim 2 divided Kukrail and Tushar cultivars, while Dim 1 segregated Pranjal and Kukrail cultivars. The score plot showed that, in contrast to N content, area per leaf, C*i*, shoot dry weight and menthol content, which contributed the least, the variables of leaves per plant, leaf area per plant, shoot height and *E* participated maximally in the differentiation of the studied cultivars of peppermint.

The PCA-biplot (Figure 14) which evaluated the separate treatments’ contribution in respective cultivars indicated that all the applied fifteen treatments of N segregate in the score plot. Nitrogen supplementation of 1.5% in all three cultivars appears separately from their control treatments. The other N supplementation treatments of 0.5%, 1.0%, and 2.0% N, on the other hand, cluster in the score plot.

The heat map analysis was used to evaluate the clustering pattern of the examined parameters in relation to N supplementation in the three peppermint studied cultivars. The N content grouped apart from the other characteristics tested, according to heat map analysis. Carbonic anhydrase activity and menthol content, on the other hand, were observed to belong to the same cluster. The K content, *E*, and *gs* were found to constitute a distinct cluster. Leaf area per plant, area per leaf and C*i* were identified to cluster together. The P*_N_*, root fresh weight, chlorophyll content and root dry weight were found to cluster together. The P content, menthol yield per plant, herbage yield per plant, EO content and EO yield were found closely associated, in a single cluster. The remaining five parameters were observed to cluster into a distinct cluster based on Euclidian distance with respect to N supplementation (Figure 15).

### 2.6. Correlation Analysis

Pearson correlation analysis was carried out at a 95% confidence level. The correlation analysis was performed among all the studied traits in all three cultivars of peppermint. In cultivar Kukrail, the correlation between menthol content and area per leaf, menthol content and N content and menthol content and EO content showed positive values, with r0.05 = 0.82, r0.05 = 0.81 and r0.05 = 0.99, respectively. Indeed, almost all the studied traits in the cultivar Kukrail showed a positive correlation, as shown in (Appendix A). Moreover, in Tushar, the correlation between menthol content and area per leaf, menthol content and N content and menthol content and EO content showed positive values, with r0.05 = 0.93, r0.05 = 0.69 and r0.05 = 0.92, respectively. The rest of the traits studied in the cultivar Tushar also showed positive correlations (Appendix A). In Pranjal, the correlation between menthol content and area per leaf, menthol content and N content and menthol content and EO content also showed positive values, with r0.05 = 0.85, r0.05 = 0.53 and r0.05 = 0.86, respectively. The other studied parameters in the cultivar Pranjal also showed a positive correlation (Appendix A).

## 3. Discussion

The current study was planned to reconnoiter the performance of peppermint cultivars by supplying different doses of N as a foliar spray. The responses of studied cultivars were studied in terms of growth, physio biochemical characteristics, yield and quality parameters. The survey of data (Figure 1 and Figure 2) showed that the N supplementation influenced all phenological traits studied. The enhancement in various growth traits studied may be attributed to the roles of N in plants. Nitrogen is a component of several metabolic compounds as well as chlorophyll, cell walls, enzymes, hormones, nucleic acids, proteins and vitamins [16]. These compounds help in cell division, cell enlargement, tissue and organ formation among others. Improved parameters for these compounds would have culminated in the enhanced shoot and root fresh weight; these in turn would have increased dry weight. Hence, higher values were observed for shoot and root fresh and dry weight. Our results resemble the findings of [17] on *Ocimum basilicum* L., [18] on *Mentha piperita* L. and *Mentha spicata* L., [19] on *Thymus vulgaris* L., [20] on *Trigonella corniculata* L.

The foliar spray of N enhanced the photosynthetic, gas exchange parameters (*P_N_*, *gs*, *E* and *Ci*), CA activity and mineral nutrient contents over water spray treatments (Figure 3, Figure 4A–C). The increase in chlorophyll content may be attributed to the roles of N in the synthesis of proteins and enzymes and the uptake of minerals including magnesium, among others [16,21,22]. These compounds, involved directly or indirectly in the biosynthesis of chlorophyll, might have enhanced the chlorophyll content of treated plants. The increase in *P_N_*, *gs* and *Ci* may be due to the role of N in the biosynthesis of RuBisCo and other photosynthetic enzymes, light-harvesting proteins and electron transport proteins, among others. The N-mediated improved synthesis of these proteins and enzymes might have contributed to enhanced photosynthetic parameters. The increase in *E* resulting from the N spray may be ascribed to its role in promoting the absorption of nutrients including K^+^ ions [23]. Potassium ions maintain the osmotic potential of cells; thus, the higher concentration of K^+^ ions would have retained more water in the cells leading to higher values for *E* in treated plants. The amelioration in CA activity in N-treated plants may be due to its role in protein synthesis including CA protein. The higher concentration of the CA enzyme would naturally be responsible for the improved activity of the enzyme. The results corroborate with the findings of [24] on *Brassica juncea* L. [25,26], on *Mentha spicata* L. and on *Mentha piperita* L. [27]. The foliar spray of N improved leaf N, P and K contents compared to the water spray treatment (Figure 4A–C). The enhancement in the leaf mineral contents may be ascribed to the role of N in the biosynthesis of cytokinins and other metabolites. Cytokinin regulates macronutrient homeostasis in plants by controlling the gene expression of transporters for nitrate, phosphate, potassium and sulphate among others [28,29]. These expressed transporters would in turn improve the absorption of mineral nutrients leading to higher values for leaf N, P and K contents of treated plants. Our observations were also endorsed by the findings of [30,31] on *Ocimum basilicum* L. and [32] on *Foeniculum vulgare* L.

The N spray improved the cell viability of root tissues and size and also the density of stomata and trichomes in peppermint plants sprayed with N (1.5%) as compared to the control (Figure 7 and Figure 8). The amelioration in cellular viability may be attributed to the roles of N in the biosynthesis of cytokinins and other metabolic compounds. Cytokinins are known to delay senescence [33], contributing to an improvement in cell viability in root tissue of N-sprayed plants. The increase in the size of stomata may be due to the roles of N in the absorption of mineral elements including K^+^ ions. Potassium ions are known to act as an osmoticum [34] leading to an improvement in the size of stomata and trichomes of N-treated plants. The augmentation in the density of stomata and trichomes and trichome diameter may be related to the roles of N in enhancing the overall growth and its direct role in the biosynthesis of cytokinins via upregulating the gene expression of adenylate isopentenyl transferase (*IPT3*) [34,35]. Thus, N would have directly or indirectly increased the density of stomata and trichomes of N-treated plants. A schematic representation drawn from this present work illustrates how the N application might have mediated the changes to improve the overall performance of peppermint cultivars (Figure 16). The results on the improvement in size and density of stomata and trichomes due to N application corroborate with the findings of [36] on *Tagetes minuta* L. and [37] on *Medicago sativa* L.

The foliar application of N enhanced the mentioned yield features of a plant (Figure 4D–F and Figure 5). The N-mediated improved plant growth and physio biochemical parameters would have contributed to comparative higher fresh matter resulting in a higher value for herbage yield per plant. The enhancement in EO content with N application may be associated with its role in the biosynthesis of many organic compounds including acetyl-CoA, amino acids, proteins and enzymes, which indeed would have played a key role in the biosynthesis of EO leading to higher values for EO of treated plants. The higher values for herbage yield and EO content would naturally correspond to EO yield per plant. The increase in menthol content of the oil due to the spraying with N may be attributed to its role in the biosynthesis of terpenoids including monoterpenes [38], resulting in a higher value for the menthol content of treated plants. The N-mediated higher values for herbage yield per plant and EO content would naturally give more EO yield per plant and EO yield per plant; the increased menthol content of the oil would give more menthol yield per plant. These results are supported by the findings of [27,39] on *Mentha piperita* L. and [25] on *Mentha spicata* L.

The 2% N application resulted in a decrease in the above-studied parameters. The reduction might be due to the decreased activity of the urease enzyme, implying that the higher dose of N application affects urease activity in plants [40]. The urease enzyme is important for N metabolism, hydrolyzing the foliar-applied urea into carbon dioxide and ammonia; thereafter, glutamine synthetase incorporates ammonia into glutamate [11]. Moreover, higher doses of N application may cause phytotoxicity which in turn lead to the plasmolysis of cells [41]. Khan et al. [41] and Otalora et al. [42] also suggested that higher doses of N application lead to phytotoxicity which in turn reduced the N uptake and decline in the yield attributes of plants.

Kukrail outperformed the other cultivars, especially after being sprayed with 1.5% N. The improved performance may be attributed to its superior genetic makeup, which would have interacted more effectively in comparison to the other cultivars. Additionally, different species or even cultivars of the same species have different nutritional needs [43]. As a result, it is possible to hypothesize that the three cultivars may have different nutritional needs and different rates of nitrogen uptake, utilization, assimilation, and remobilization, all of which may have contributed to the considerable differences between the three cultivars. The findings regarding the varying performance of peppermint cultivars resemble those of [44] on *Mentha piperita* L. and [44] and [45] on *Mentha arvensis* L.

## 4. Materials and Methods

### 4.1. Collection and Establishment of Plants

The healthy and fresh runners of various peppermint cultivars were procured from the Central Institute of Medicinal and Aromatic Plants, Lucknow (India) against proper receipts. They were planted in a nursery of the department. The nursery was properly irrigated and maintained for 20 days. A preliminary experiment was conducted to assess various growth-related traits and physio biochemical processes of peppermint cultivars. Based on these parameters, the best three superior cultivars, Kukrail, Pranjal and Tushar were selected for further experimental purposes.

### 4.2. Experimental Setup and Treatment Design

For the current study, a randomized experiment was laid down in earthen pots in the Botany Department, Aligarh Muslim University, Aligarh India (27°54′25.91" N, 78°4′31.48" E and 192.92 m altitude) under natural environmental conditions. Earthen pots were prepared uniformly by filling them with a 5 kg mixture of farmyard manure and sandy loam soil in a proportion of 1:4. Before transplanting the peppermint runners, soil samples were collected from various pots for sample preparation for testing the soil characteristics. The available nutrients, viz., N, P and K, were present at the levels of 96.4, 8.7 and 145.1 mg kg^−1^ soil, respectively, at the time of transplantation. From the nursery, three fresh and healthy runners were transplanted into each pot. Prior to transplanting, an appropriate dose of N, P and K was uniformly supplied to the soil according to the procedure of Reddi and Reddy [46]. The N, P and K were supplied in the form of CO(NH_2_)_2_, P_2_O_5_ and K_2_O, respectively. During our study, sixty pots were prepared and arranged into three groups each of twenty pots with each group of pots allocated to separate cultivars. The leaves of the plants were sprayed with N at 0 (water), 0.5, 1.0, 1.5 and 2% N twice with the first spray treatment being given at 60 DAT (spray time 9:30 am, average temperature 28 °C, relative humidity 32% and wind speed 11.3 mph) and the second spray treatment at 80 DAT (spray time 9:30 am, average temperature 33 °C, relative humidity 28% and wind speed 10.6 mph). Morphological, physio biochemical and microscopical observations were conducted at 100 DAT and yield parameters at 120 DAT. The representative pot images of our experiment are shown in Figure 17A–C.

### 4.3. Determination of Plant Growth Characteristics

To analyze the morphological parameters of peppermint cultivars, the full-grown runners were uprooted carefully from the earthen pots and cleaned with tap water to remove the soil from the roots. The measurement of the shoot and root length of plants was recorded with a measuring scale and presented in cm. The area of a leaf was taken with the help of graph paper and designated in cm^2^, while the leaves of a plant were computed manually. The leaf area of the plant was calculated on the basis of the leaves per plant and the area per leaf. The shoot and root fresh weight was recorded with electronic analytical balance and expressed in g. For dry weight analysis, plant samples were kept in an oven overnight at 80 °C till a constant weight was attained.

### 4.4. Physio Biochemical Parameters

#### 4.4.1. Chlorophyll Content

On a bright sunny day at 11 a.m., the chlorophyll content of undamaged, whole leaves was measured with a chlorophyll meter SPAD-502 (KMS Inc., Tokyo, Japan).

#### 4.4.2. Gas Exchange Parameters

The gas exchange parameters were evaluated with an Infrared Gas Analyzer (IRGA) (LI-COR-6400, Lincoln, NE, USA). The fully grown leaves of the main plant were chosen to note the readings of intercellular CO_2_ concentration (*C_i_*), net photosynthetic rate (*P_N_*), stomatal conductance (*g_s_*) and transpiration rate (*E*). After adjusting the leaf chamber, the readings of the above parameters were observed within a minute at 11:00 am under optimum environmental conditions.

#### 4.4.3. Carbonic Anhydrase Activity

The method of Dwivedi and Randhawa [47] was used to calculate the carbonic anhydrase (CA) activity. Small portions of fresh young leaves were placed in Petri dishes with a 10 mL aqueous cysteine hydrochloride solution (0.2 M). After incubating the leaves for 20 min at 4 °C, the solution was blotted off the leaves. The fragments were placed in a reaction vessel, and phosphate buffer (pH 6.8), 0.2 M NaHCO_3_, and 0.002% bromothymol blue indicator were added. The reaction vessel was stirred and then left at 4 °C for thirty minutes. Thereafter, methyl red was used to titrate the reaction content in presence of 0.05 N HCl. Finally, µ mol CO_2_ kg^−1^ FWs^−1^ was used to quantify the enzyme’s activity.

#### 4.4.4. Leaf Nutrient Content (N, P and K)

Paper-wrapped leaf samples were dried at 80 °C in an oven for 48 h. After drying, these leaves were mortar-pounded into a fine powder. The 100 mg leaf powder was placed in a test tube containing 2 mL H_2_SO_4_. After heating the test tube at 80 °C for 2 h, the reaction mixture was allowed to cool down for 15 min. After cooling the mixture, 0.5 mL of 30% H_2_O_2_ was added dropwise and heated at 50 °C until the solution turned bright yellow. The heating and cooling process and the addition of H_2_O_2_ were continued till the colour of the reaction content become colourless. Finally, this reaction content was taken for the evaluation of leaf N, P and K contents. The leaf N and P contents were evaluated by adopting the methods of Lindner [48] and Fiske [49], respectively. A flame-photometer (Model: C150, AIMIL, India) was used to estimate the K content by following the protocol of Hald [50].

### 4.5. Microscopic Examination

The microscopical studies were performed in the samples of peppermint cultivars treated with 0% and 1.5% N.

#### 4.5.1. Confocal Microscopy

The thin longitudinal samples from the root cuttings of three cultivars were stained with propidium iodide for twenty minutes before being examined under a microscope. The roots were previously cleaned twice in distilled water. After being stained, the root samples were put on a glass slide and studied by using a confocal laser scanning microscope (LSM 780, ZEISS, Jena, Germany).

#### 4.5.2. Scanning Electron Microscopical Analysis

Scanning Electron Microscopical analyses (SEM) (JEOL, JSM 6510 LV, Akishima, Japan) were performed in order to visualize the trichome and stomatal appearance on the leaves of the three studied cultivars. Leaf samples were preserved in 2 percent formaldehyde (1 mL), 2.5 percent glutaraldehyde (98 mL) and 100 mM sodium cacodylate (1 mL). For maintaining the pH of the solution, phosphate buffer (100 mM, pH 7.3) was added dropwise and left for two hours. Ethanol grades (30%, 50%, 70%, and 100%) were used to dehydrate the leaf samples. The dehydrated leaf samples were then cut into small strips of about 2 cm^2^ and besmeared with gold-palladium. Stomatal behavior was recorded at 500× and 3000× and trichome external appearance was examined at 100 and 1000× at 15 kV. The size and density of both stomata and trichomes were measured in μm and mm^2^, respectively.

### 4.6. Yield and Quality Attributes

At the maturation stage (120 DAT), characteristics such as herbage yield, EO content, EO yield, menthol content and yield were evaluated.

#### 4.6.1. Essential Oil Extraction of Peppermint Cultivars

Clevenger’s apparatus was used to extract peppermint EO. Leaves of about 50 g were taken from plants and cut into small pieces and then placed in a flask attached to Clevenger’s apparatus condenser containing 500 mL of distilled water. The hydrodistillation process lasted three hours. Following that, the EO content (percentage) and EO yield were computed. Thereafter, oven-dried anhydrous sodium sulphate was used to dry the extracted oil, which was then maintained at 4 °C for GC-MS analysis.

#### 4.6.2. GC-MS Analysis of EO

A GC-MS-TQ8050 NX (Shimadzu Corporation, Kyoto, Japan) was used to analyze the EO of the three cultivars at the Central Instrumentation Laboratory of the Central University of Punjab in Bathinda, India. With a split ratio of 5, the split injector temperature was set at 280 °C. The oven’s temperature was programmed from 0 to 40 °C and then increased to 220 °C with a hold period of three minutes and five minutes, respectively. Thereafter, the temperature was further increased to 250 °C and held for 5 min. Helium was used as the carrier gas, flowing at a constant rate of 1 mL min^−1^, with the ion source temperature set at 230 °C, interface temperature of 250 °C and a mass scan range of 40–800 amu. The column flow was set at 1.00 mL min^−1^, while the column oven temperature was set at 40 °C. The volume used for sample loading and injection was 1 µL. Using the NIST17R library and NIST17M2 library, the various bioactive compounds of the EO were analyzed based on m/z ratio and their retention time. The menthol content in the control and treated plants was estimated by calculating the peak area of the chromatogram.

### 4.7. Statistical Analysis

The SPSS 25.0 statistical software (SPSS Inc., Chicago, IL, USA) was used to analyze the data and was expressed as mean ± SE (Standard Error). The variance and significance in the data were evaluated by Duncan’s Multiple Range Test (DMRT) and Tukey’s test at *p* ≤ 0.05. Bar plots were made using MS Excel. The line plots, heat map, principal component analysis and Pearson correlation were performed using “R” statistical software. The ImageJ software (version 19.00) was used to analyze the SEM images.

## 5. Conclusions

The results of this study showed that the three peppermint cultivars studied for growth, productivity, and EO content (especially menthol) after receiving foliar applications of N performed better. However, the three cultivars benefited most from a 1.5% N spray, while 2% N application showed a lesser positive effect. Among the three, the Kukrail cultivar performed the best, followed by the Tushar and the Pranjal. Based on the results of the current research, it is suggested that a 1.5% N treatment rate is optimal for enhancing peppermint crop yield, leaf nutrient content, biochemical and qualitative characteristics. However, more research is needed to determine how N contributes to the biosynthesis of secondary metabolites in peppermint plants.

## Figures and Tables

**Figure 1 plants-12-00809-f001:**
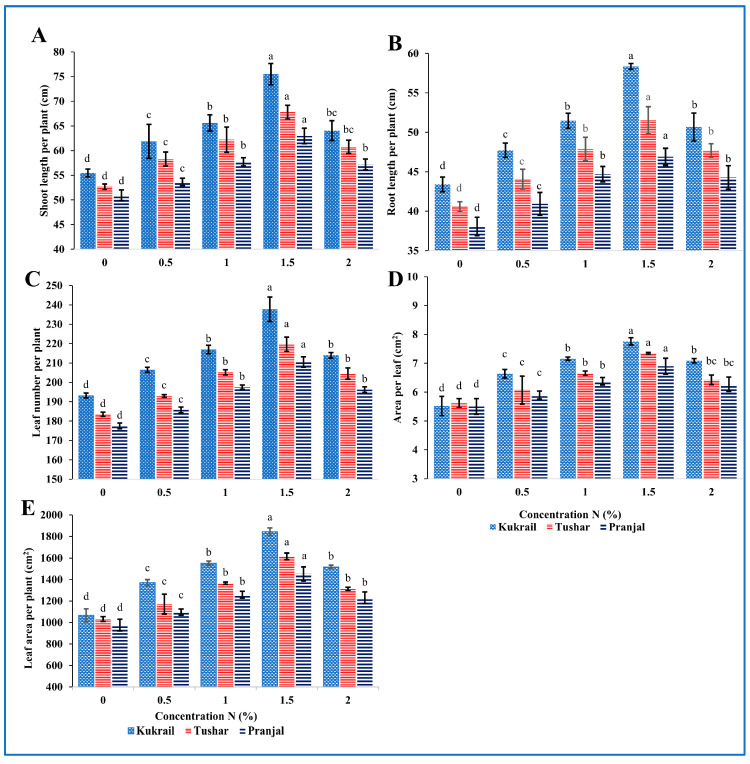
Effect of N on three cultivars: (**A**) Shoot length per plant, (**B**) Root length per plant, (**C**) Leaf number per plant, (**D**) Area per leaf, (**E**) Leaf area per plant of the three peppermint cultivars. Data represent the mean and bars represent the standard error (SE) of four replicates (*n* = 4) of each cultivar. The letters over bars represent significance at *p* ≤ 0.05 by Duncan’s Multiple Range Test (DMRT).

**Figure 2 plants-12-00809-f002:**
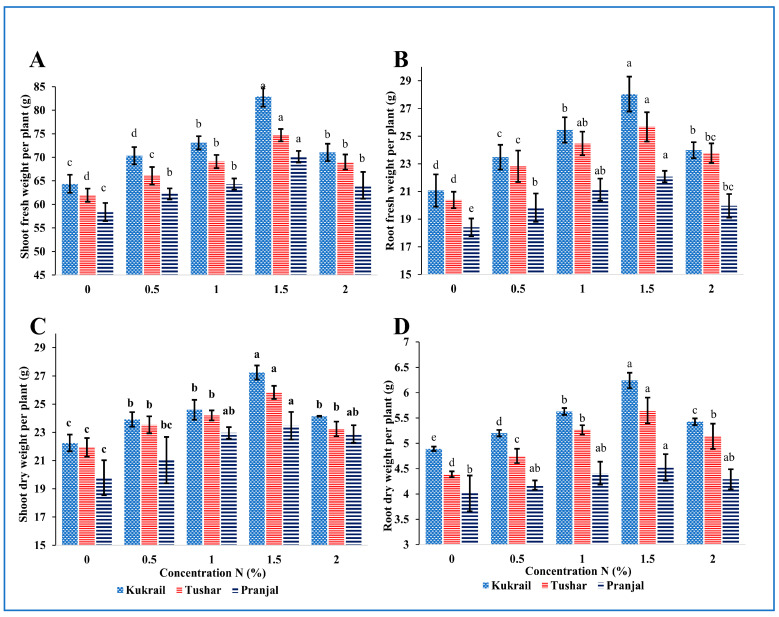
Effect of N on three cultivars: (**A**) Shoot fresh weight per plant, (**B**) Root fresh weight per plant, (**C**) Shoot dry weight per plant, (**D**) Root dry weight per plant of the three peppermint cultivars. Data represent the mean and bars represent the standard error (SE) of four replicates (*n* = 4) of each cultivar. The letters over bars represent significance at *p* ≤ 0.05 by Duncan’s Multiple Range Test (DMRT).

**Figure 3 plants-12-00809-f003:**
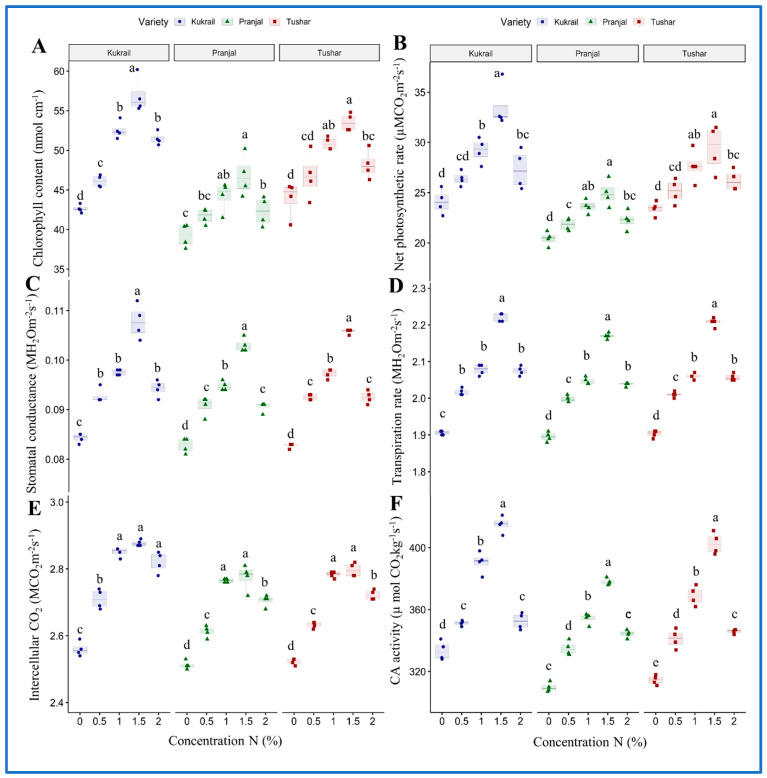
Effect of N on three cultivars: (**A**) Chlorophyll content, (**B**) Net photosynthetic rate, (**C**) Stomatal conductance, (**D**) Transpiration rate, (**E**) Intercellular CO_2_ concentration, (**F**) CA activity of the three peppermint cultivars. The distinguished colored shapes in the box plot represent the individual reading of four replicates of each cultivar. The letters over the shapes represent the significance at *p* ≤ 0.05 by Duncan’s Multiple Range Test (DMRT).

**Figure 4 plants-12-00809-f004:**
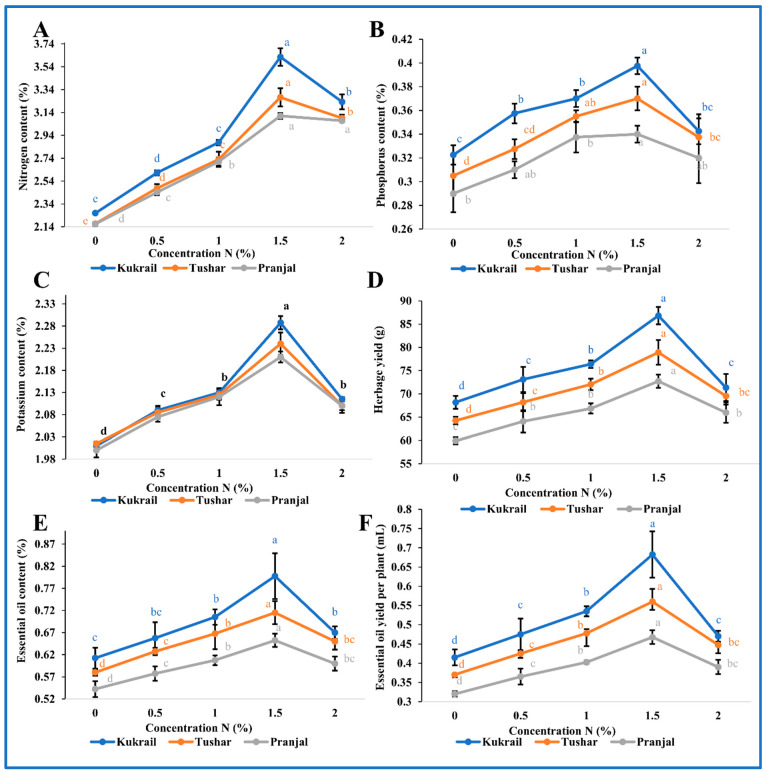
Effect of N on three cultivars: (**A**) Nitrogen (%) of leaves (**B**) Phosphorus (%) of leaves (**C**) Potassium (%) of leaves (**D**) Herbage yield (**E**) Essential oil content (**F**) Essential oil yield per plant of the three peppermint cultivars. Lines represent the mean and bars represent the standard error (SE) of four replicates *(n* = 4) of each cultivar. The letters represent significance at *p* ≤ 0.05 by Duncan’s Multiple Range Test (DMRT).

**Figure 5 plants-12-00809-f005:**
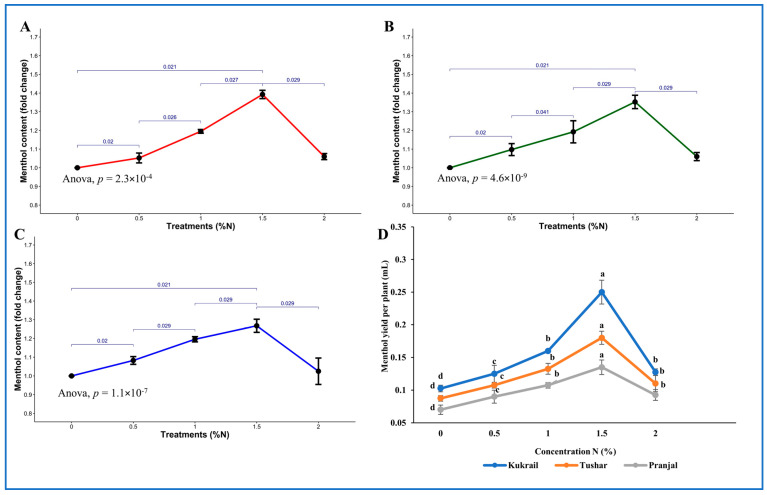
Effect of N on the menthol content of three cultivars. (**A**) Kukrail, (**B**) Tushar, (**C**) Pranjal, (**D**) Effect of N on the menthol yield of three peppermint cultivars.

**Figure 6 plants-12-00809-f006:**
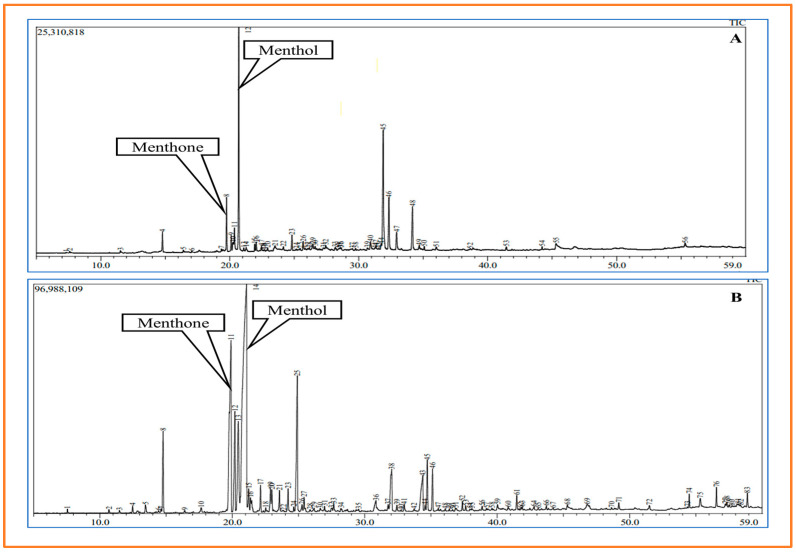
Chromatogram of EO content obtained from the leaves of peppermint (cultivar Kukrail). (**A**) Showing peak of menthol in control (peak 12), (**B**) Showing peak of menthol at 1.5% N (peak 14).

**Figure 7 plants-12-00809-f007:**
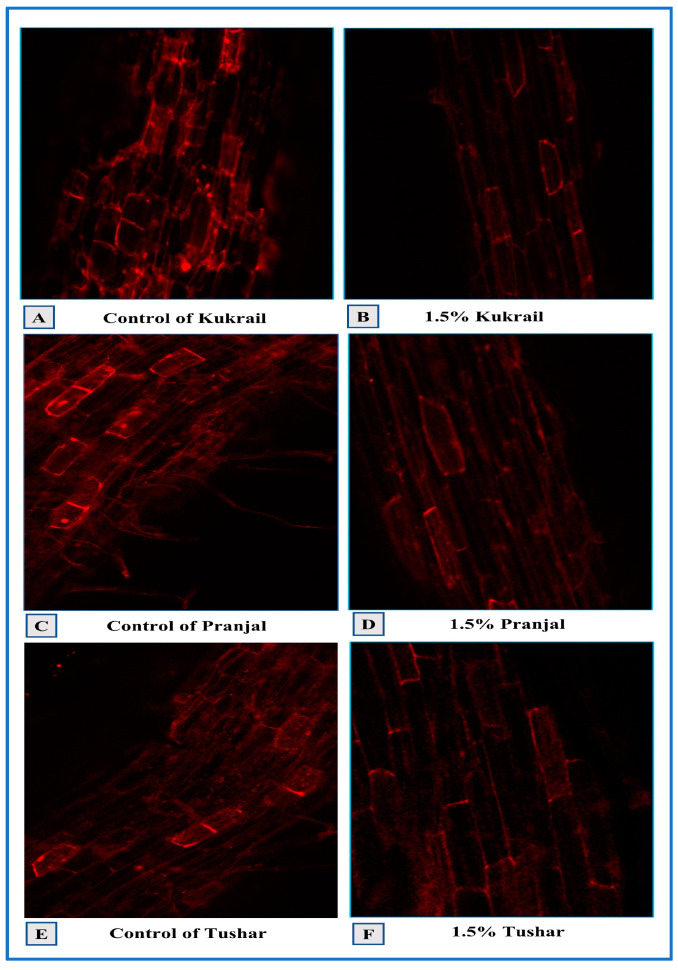
Confocal microscopy depicting the cellular viability of peppermint cultivars. The cellular viability in control plants showed higher fluorescence confirming a large number of dead cells. In contrast, cellular viability at 1.5% N having the lesser fluorescence depicts the enhanced cellular viability in the studied cultivars. (**A**) control of Kukrail, (**B**) 1.5% Kukrail, (**C**) control of Pranjal, (**D**) 1.5% Pranjal, (**E**) control of Tushar, (**F**) 1.5% Tushar.

**Figure 8 plants-12-00809-f008:**
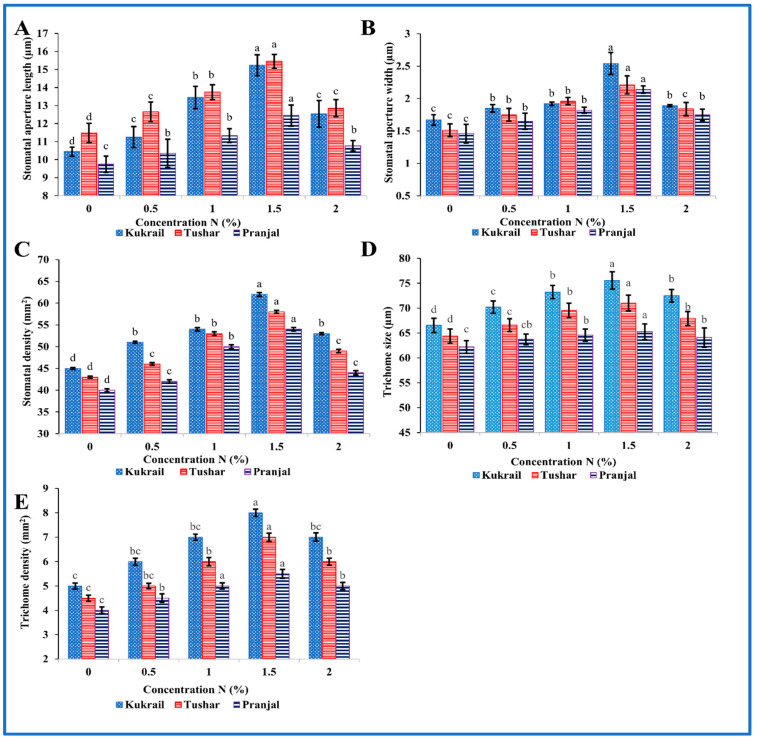
Effect of N on three cultivars: (**A**) Stomatal aperture length, (**B**) Stomatal aperture width, (**C**) Stomatal density, (**D**) Trichome size, (**E**) Trichome density of the three studied peppermint cultivars. Data represent the mean of four replicates (*n* = 4) and bars represent their standard error (SE). The letters over bars represent significance at *p* ≤ 0.05 by Duncan’s Multiple Range Test (DMRT).

**Figure 9 plants-12-00809-f009:**
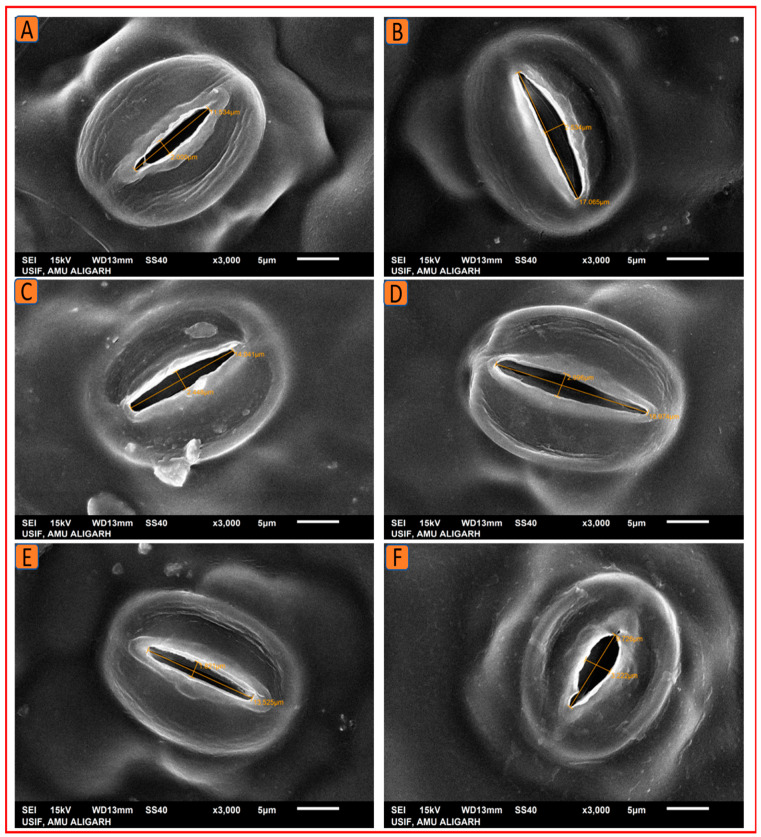
SEM micrographs depicting the stomatal behavior of the three studied peppermint cultivars. The SEM micrographs confirmed an increase in the stomatal dimensions in all three cultivars which had received the N application. (**A**) Stomatal aperture size at 0% N in Kukrail, (**B**) at 1.5% N in Kukrail, (**C**) at 0% N in Pranjal, (**D**) at 1.5% N in Pranjal, (**E**) at 0% N in Tushar, (**F**) at 1.5% N in Tushar.

**Figure 10 plants-12-00809-f010:**
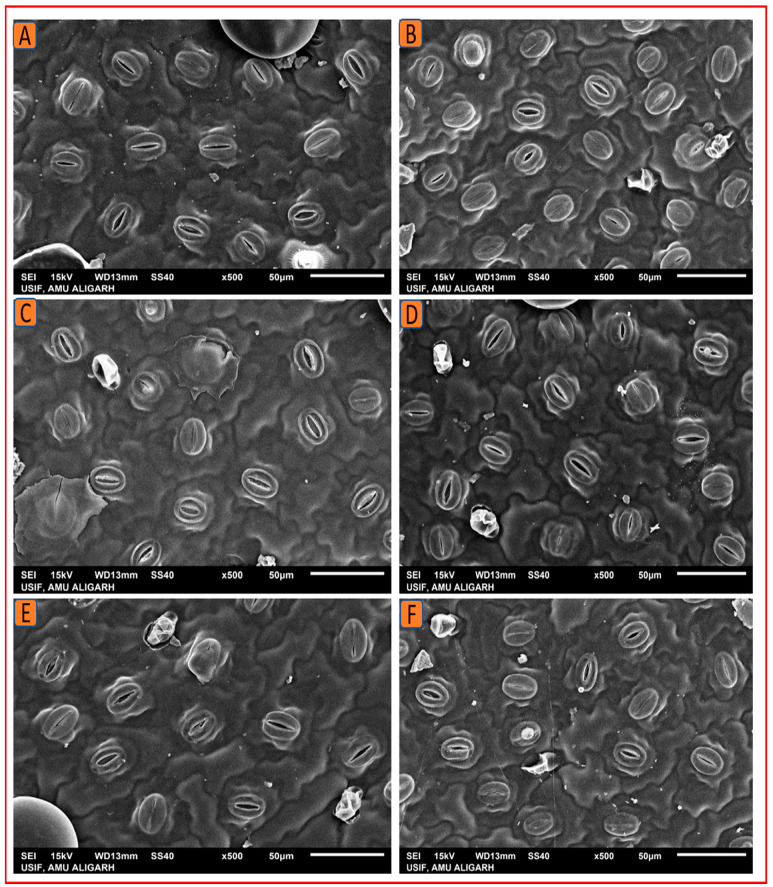
SEM micrographs depicting the stomatal density of the three peppermint cultivars studied. (**A**) Stomatal aperture density at 0% N in Kukrail, (**B**) at 1.5% N in Kukrail, (**C**) at 0% N in Pranjal, (**D**) at 1.5% N in Pranjal, (**E**) at 0% N in Tushar, (**F**) at 1.5% N in Tushar.

**Figure 11 plants-12-00809-f011:**
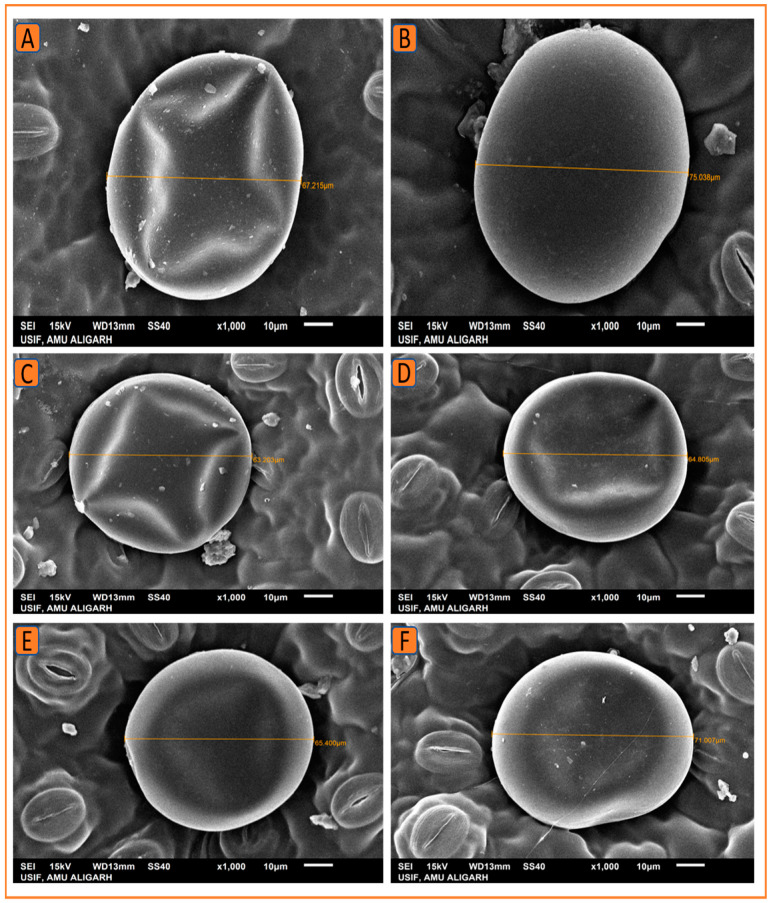
SEM micrographs showing the trichome size in the three studied cultivars of peppermint. The SEM micrographs show an enhancement in trichome size in all three cultivars by N application. (**A**) Trichome size at 0% N in Kukrail, (**B**) at 1.5% N in Kukrail, (**C**) at 0% N in Pranjal, (**D**) at 1.5% N in Pranjal, (**E**) at 0% N in Tushar, (**F**) at 1.5% N in Tushar.

**Figure 12 plants-12-00809-f012:**
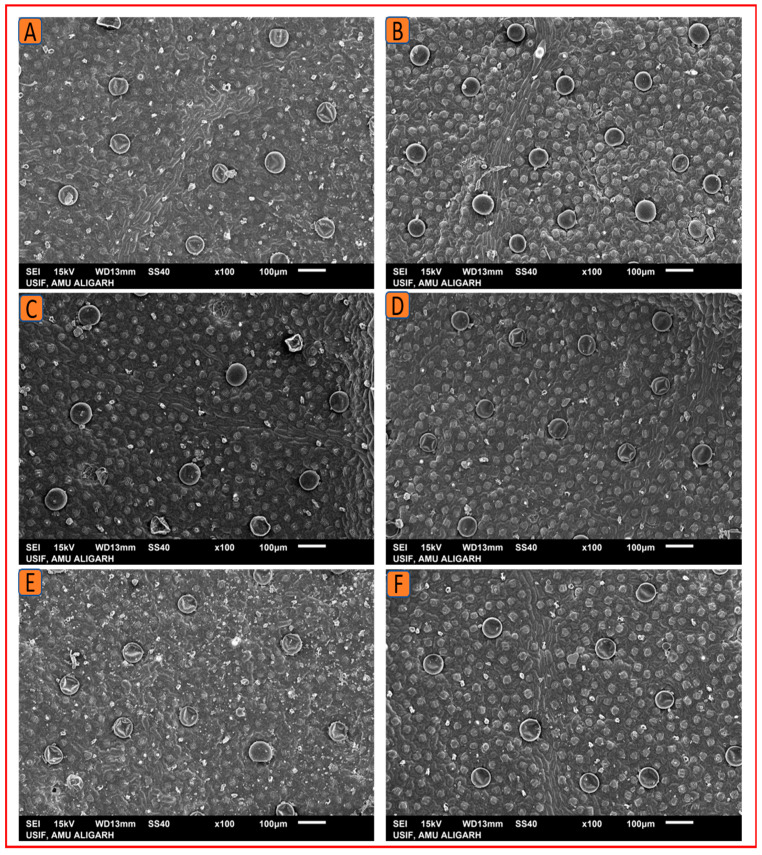
SEM micrographs showing trichome density of in the three cultivars of peppermint. The SEM micrographs show an enhancement in the trichome density by N application. (**A**) Trichome density at 0% N in Kukrail, (**B**) at 1.5% N in Kukrail, (**C**) at 0% N in Pranjal, (**D**) at 1.5% N in Pranjal, (**E**) at 0% N in Tushar, (**F**) at 1.5% N in Tushar.

**Figure 13 plants-12-00809-f013:**
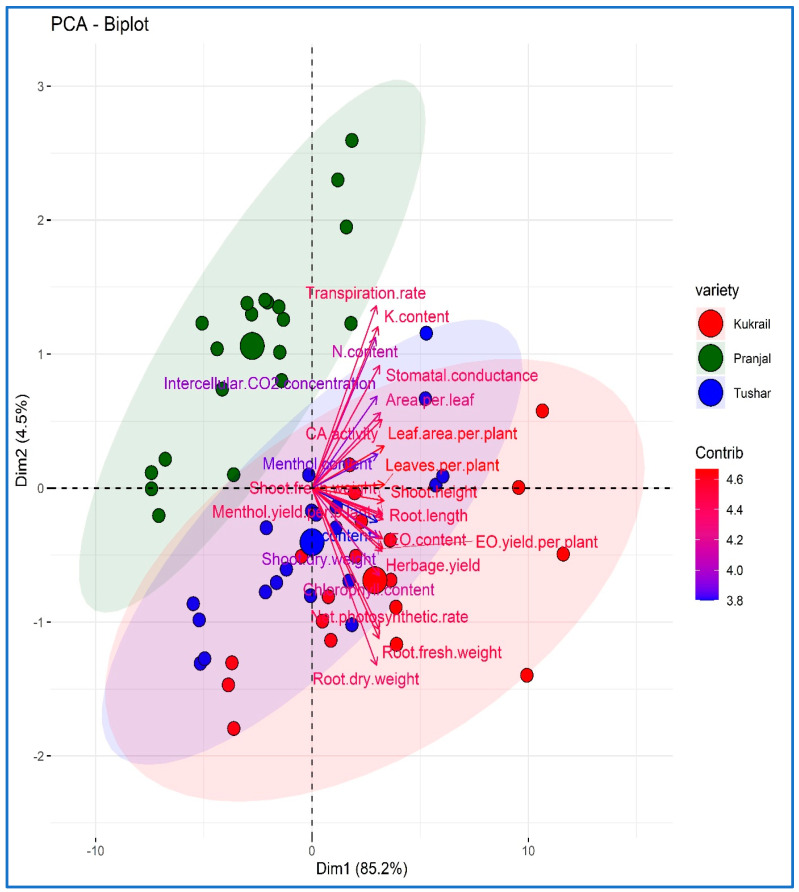
PCA-biplot depicting the effect of N on the performance of three studied cultivars of peppermint.

**Figure 14 plants-12-00809-f014:**
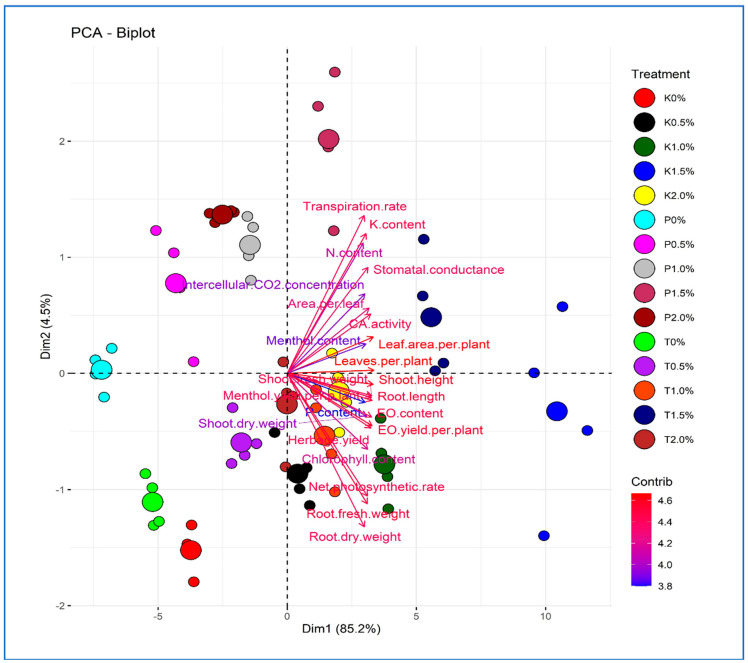
PCA-biplot depicting the interaction effect of three cultivars and different N treatments. K0%–K2%: cultivar Kukrail at different concentrations of P0%–P2%; cultivar Pranjal at different concentrations of T0%–T2%; cultivar Tushar at different concentrations of %N.

**Figure 15 plants-12-00809-f015:**
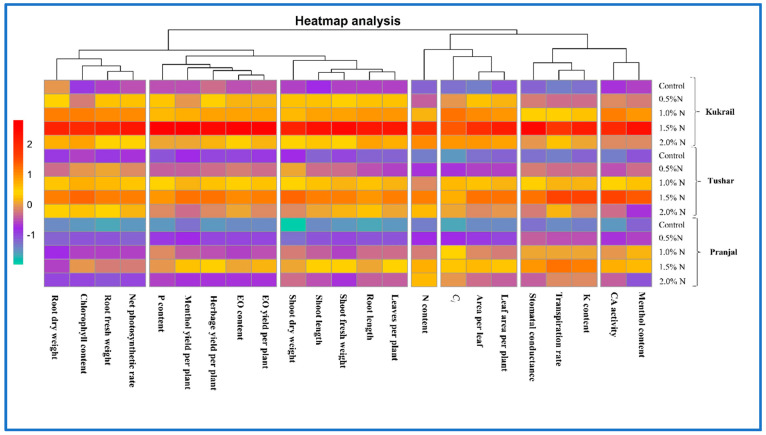
Heat map analysis of the studied parameters of three cultivars of peppermint.

**Figure 16 plants-12-00809-f016:**
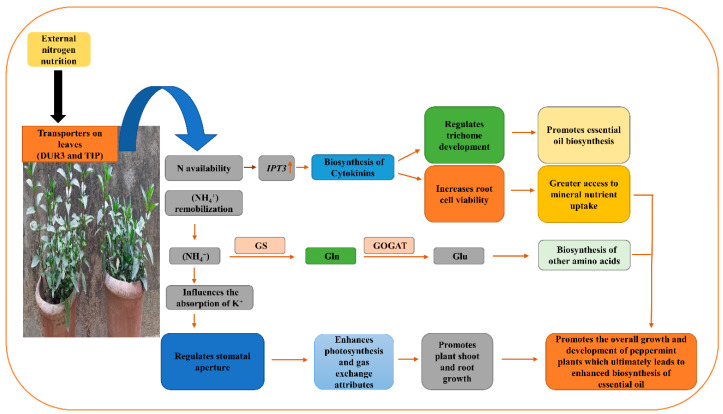
A schematic representation showing the potential mechanism of N application in improving the performance of peppermint plants. *IPT3*: adenylate isopentenyl transferase; GS: Glutamine synthetase; GOGAT: Glutamine oxoglutarate aminotransferase; Gln: Glutamine; Glu: Glutamate. Upward arrow indicates upregulation.

**Figure 17 plants-12-00809-f017:**
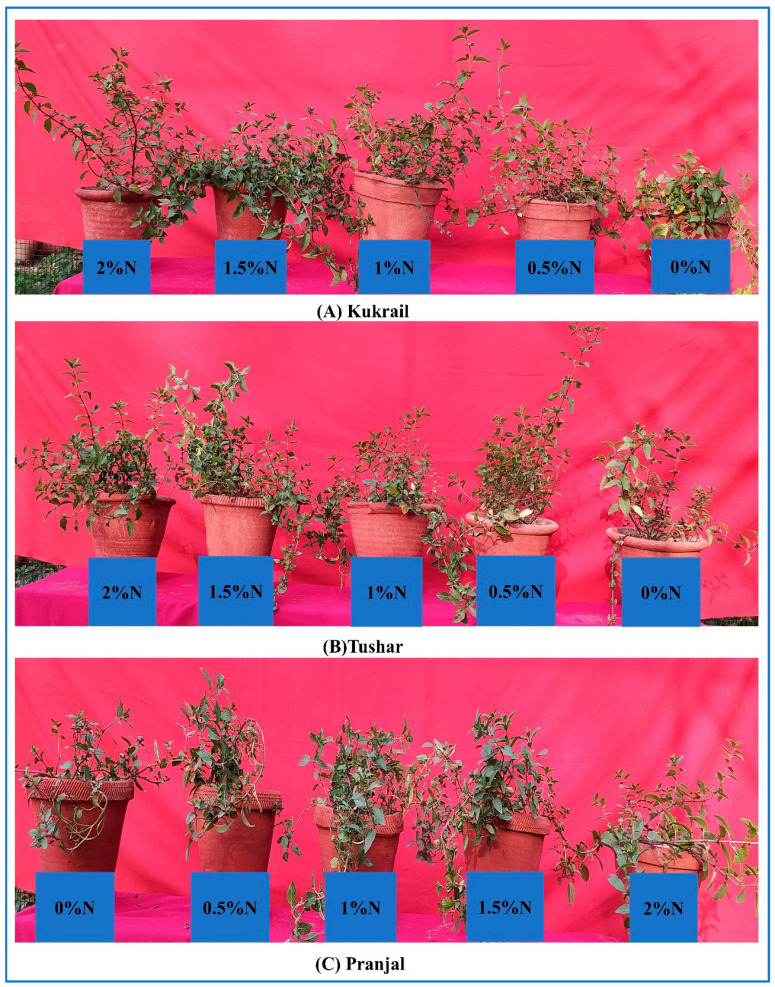
Representative experimental plants of the three cultivars of peppermint.

## Data Availability

All data are available on in this publication.

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
