# Peer review of "Nitrogen Supplementation Modulates Morphological, Biochemical, Yield and Quality Attributes of Peppermint"

_plants, 2023, doi:10.3390/plants12040809_

Round 1

Reviewer 1 Report

This manuscript evaluated effect of nitrogen on physiobiochemistry of three different peppermint cultivars. Authors comprehensively compared factors regarding various growth factors, cellular viability, leaf morphology and essential oils. Overall, the study is very interesting however some of the experimental procedure and results should be explained more detail before further considerations.

Here are some of my points.

-In this manuscript, authors selected three peppermint cultivars and compared.  Please describe why authors selected these three specific cultivars (Kukrail, Pranjal and Tushar) from peppermint and also should mention in discussion part why there was significant differences between three cultivars.

-Every result showed that 1.5% of nitrogen was optimal condition while higher than 1.5% showed less effective. Please describe reason of this phenomenon in discussion part and compare with other conducted studies which evaluated effect of nitrogen.

-Regarding analysis of essential oils, although chromatogram of EO extracts showed many peaks, it seems that authors only quantified menthol from EO extracts. How about other essential oil compositions? If authors only quantify the menthol from essential oils, authors should mention in the manuscript as quantification of “menthol” not “essential oils”.

-In figure 8, authors compared viability of cells using propidium iodide with confocal microscope. By just looking at Figure 8, it is not easy to compare with exact criteria. It seems that Figure 8C seems better than 8B in my opinion, although authors mentioned 8B was the best. So please describe in detail, how authors compared viabilities with Figure 8. (Which parameters or criteria is applied for the comparison?)

Minor points

-Please consider rephrasing the title. Current title focused too much on nitrogen.

-In section 2.6.2 about GC-MS analysis methods, GC column information is missing.

-What is criteria regarding Figure 2 and Figure 3? Figure 2F may fits better when position in Figure 3 which the contents are about weight composition.

-Page 12 Line 290, (Figure) à should mention which figure.

-Page 14 Line 327, 15.24μm à 15.24 μm  (Space between number and unit)

Author Response

Rebuttal letter

Journal title: Plants

MS id: plants2201665

Dear Editor,

We are extremely grateful to the reviewers for their valuable comments, suggestions and corrections, which have improved our manuscript to a great extent. It is hoped that the revised MS will meet to the level of their satisfaction.

Reviewer 1

Comments

*In this manuscript, authors selected three peppermint cultivars and compared. Please describe why authors selected these three specific cultivars (Kukrail, Pranjal and Tushar) from peppermint and also should mention in discussion part why there was significant differences between three cultivars.

  • Response

It has been justified in the revised MS.

*Every result showed that 1.5% of nitrogen was optimal condition while higher than 1.5% showed less effective. Please describe reason of this phenomenon in discussion part and compare with other conducted studies which evaluated effect of nitrogen.

  • Response

It has been elaborated in the revised MS.

*Regarding analysis of essential oils, although chromatogram of EO extracts showed many peaks, it seems that authors only quantified menthol from EO extracts. How about other essential oil compositions. If authors only quantify the menthol from essential oils, authors should mention in the manuscript as quantification of “menthol” not “essential oils”.

  • Response

The suggested correction has been made in the revised MS.

*In figure 8, authors compared viability of cells using propidium iodide with confocal microscope. By just looking at Figure 8, it is not easy to compare with exact criteria. It seems that Figure 8C seems better than 8B in my opinion, although authors mentioned 8B was the best. So please describe in detail, how authors compared viabilities with Figure 8. (Which parameter or criteria is applied for the comparison)

  • Response

The revised Figure has been incorporated in MS. Moreover, the control of each cultivar was compared with treatments. Furthermore, the comparison was also made within the cultivars. The comparison was carried out on the basis of fluorescence spots. 

*Please consider rephrasing the title. Current title focussed too much on nitrogen.

  • Response

The title has been modified.

*In section 2.6.2 about GC-MS analysis methods, GC column information is missing.

  • Response

It has been incorporated in the revised MS.

*What is criteria regarding Figure 2 and Figure 3. Figure 2F may fits better when position in Figure 3 which the contents are about weight composition.

  • Response

There are not any specific criteria for Figure 2 and Figure 3. The figures of the studied parameters are arranged in chronological order.

*Page 12 Line 290, (Figure) should mention which figure.

  • Response

It has been added.

*Page 14 Line 327, 15.24 µm (space between number and unit)

  • Response

It has been rectified.

Reviewer 2 Report

The article is devoted to the study of the effect of the treatment of leaves of mint plants with a nitrogen-containing substance on growth, morphophysiological and biochemical parameters, it is shown that treatment with 1.5% N is optimal.

However, the presented materials raise a number of questions:

1. what is the reason for the chosen spraying interval 30 and 60 days after planting the seedlings?

2. Was the initial nitrogen content of the soil in which the plants grew during the experiment taken into account?

3. Were the time of day and climatic factors (light, temperature, humidity) taken into account when spraying was carried out? Over such a long period of the experiment, environmental factors most likely could change significantly

4. How were wilted leaves taken into account when calculating their area? The technique for calculating leaf area can be improved by using a scanner and related software.

5. According to Fig. 7, it is impossible to assess the effect of treatment with a nitrogen solution on the menthol content, there is no peak intensity scale or a clearly marked area of the peaks. Why does the menthone in Fig. 7A elute at 19.8 minutes, and at 15 minutes at 7B? 

Author Response

Rebuttal letter

Journal title: Plants

MS id: plants2201665

Dear Editor,

We are extremely grateful to the reviewers for their valuable comments, suggestions and corrections, which have improved our manuscript to a great extent. It is hoped that the revised MS will meet to the level of their satisfaction.

Reviewer 2

Comments

*What is the reason for the chosen spraying interval 30 and 60 days after planting the seedlings.

  • Response:

The spray interval of 60 and 80 DAT was chosen as per our previous work of lab mate mentioned in the doctoral thesis. Moreover, these are the most crucial growth stages for foliar spray on the performance of peppermint plants.

*Was the initial nitrogen content of the soil in which the plants grew during the experiment taken into account?

  • Response

It has now been incorporated in the revised MS.

*Were the time of day and climatic factors (light, temperature, humidity) taken into account when spraying was carried out? Over such a long period of experiment, environmental factors most likely could change significantly.

  • Response

Yes, these factors were taken into consideration and have been incorporated in the revised MS.

*How were wilted leaves taken into account when calculating their area? The technique for calculating leaf can be improved by using a scanner and related software.

  • Response

As far as our experiment was conducted, the plants were watered uniformly as per their requirement in order to prevent the water deficit in the soil. During the calculation of the leaf area, the fresh and fully expanded leaves were carefully taken into account as there was hardly any wilting under our observation.

*According to Fig. 7, it is impossible to assess the effect of treatment with a nitrogen solution on the menthol content, there is no peak intensity scale or a clearly marked area of the peaks. Why does the menthone in Fig. 7A elute at 19.8 minutes, and at 15 minutes at 7B.

  • Response

We are very thankful to the reviewer for correcting us, menthone eluted at the same time in both the samples. It was a typing error before placing the menthone on the wrong peak. Now the revised figure has been incorporated in the revised MS. We also mentioned the peak number of menthol in the figure legend.

With regards

Prof. Firoz Mohammad

Department of Botany

Aligarh Muslim University

Aligarh-202002

India

[email protected]

Round 2

Reviewer 2 Report

The corrected version of the article may be published in the journal